# Assessment of Recent Changes in Dust over South Asia Using RegCM4 Regional Climate Model

**Acharya Asutosh [1],\*, S.K Pandey [1], V Vinoj [1], Ramakrishna Ramisetty [2] and Nishant Mittal [2]**

[1] Indian Institute of Technology Bhubaneswar, Bhubaneswar 752050, Odisha, India; sp25@iitbbs.ac.in (S.K.P.); vinoj@iitbbs.ac.in (V.V.)

[2] TSI Instruments India Private Limited, Bangalore 560102, Karnataka, India; rama.ramisetty@tsi.com (R.R.); nishant.mittal@tsi.com (N.M.)

\* Correspondence: aa14@iitbbs.ac.in

**Abstract:** Pre-monsoon dust aerosols over Indian regions are closely linked to the monsoon dynamics and Indian summer monsoon rainfall. Past observational studies have shown a decline in dust loading over the Indian landmass potentially caused by changing rainfall patterns over the desert regions. Such changes are expected to have a far-reaching impact on regional energy balance and monsoon rainfall. Using a regional climate-chemistry model, RegCM4.5, with an updated land module, we have simulated the long-term (2001–2015) changes in dust over the arid and semi-arid dust source regions of the North-Western part of the subcontinent. It is found that the area-averaged dust aerosol optical depth (AOD) over the arid and semi-arid desert regions has declined by 17% since the start of this millennium. The rainfall over these regions exhibits a positive trend of 0.1 mm day$^{-1}$year$^{-1}$ and a net increase of >50%. The wet deposition is found to be dominant and ~five-fold larger in magnitude over dry deposition and exhibits total changes of ~79 and 48% in the trends in atmospheric dust. As a response, a significant difference in the surface (11%), top of the atmosphere radiative forcing (7%), and widespread atmospheric cooling are observed in the short wave domain of radiation spectrum over the Northern part of the Indian landmass. Such quantification and long-term change studies are necessary for understanding regional climate change and the water cycle.

**Keywords:** dust aerosols; radiative forcing; regional climate; rainfall; RegCM

## 1. Introduction

Mineral dust (or dust) is one of the most important sources for aerosol mass in the atmosphere [1]. They are of various size ranges and exert a radiative effect in both the solar and terrestrial radiation known as the aerosol direct effect [2–4]. Based on their hygroscopic nature and secondary mixing, the dust aerosols also have an impact on the radiation by affecting cloud microphysics, known as the indirect effect [5–7]. Dust is known to have a strong influence on both regional and global climate [8,9].

The abundance of mineral dust over India makes the overall aerosol burden about three times higher than the global average, especially during summer and monsoon seasons [10]. The primary sources of dust over the Indian landmass are the arid and semi-arid regions (mostly North-Western and adjacent parts of India) and long-range transport from Middle-West Asia and North Africa by the westerly winds [11–16]. It peaks during April/May and declines thereafter due to the arrival of monsoon and enhanced wet deposition [13,15].

The direct and indirect effects take place simultaneously, interacting with each other. This adds more complexity to the overall forcing and responses of the hydrological cycle over the South Asian/Indian monsoon system [8,17]. Dust is known to alter atmospheric dynamics through warming, thereby impacting atmospheric circulation [12,18,19].

Many studies highlight the short- and long-term impacts of remote/local aerosols to monsoon rainfall through elevated heat pump (EHP) mechanism solar dimming and local and remote atmospheric heating processes [11,12,15,20–23]. The presence of absorbing aerosols such as dust and black carbon over parts of Northern India and the Himalayan foothills could enhance monsoon rainfall in subsequent months (June/July) through dynamical feedback [13,24]. Therefore, variations in the summer dust cycle could inevitably impact the monsoon hydrology by altering the regional radiation balances, thereby atmospheric dynamics.

Limited studies have highlighted the pre-monsoon dust changes over the arid and semi-arid (dust sources) regions of India [25]. In recent times, Indo-Pak deserts have gone through significant changes due to increased human activity, such as various infrastructure projects, construction activity, and increased irrigation, etc. Rainfall over the Indian subcontinent has also changed drastically over time. A previous study by Pandey et al. [25] using multiplatform satellite and ground-based observations along with reanalysis data has conclusively shown a 10–20% decrease in dust due to an increase in rainfall and a change in wind circulation. Their study attempted to investigate the long-term aerosol/dust dynamics during the pre-monsoon season over India. Further, Jin et al., 2017 [12] have also pointed out the increase in surface greenness and soil moisture over the North-Western Indian subcontinent thereby reducing local dust emission. It is to be noted that both of these studies used multiplatform satellite and ground-based observations, along with reanalysis data, and showed a 10–20% decrease in dust due to the increase in rainfall and slowing down wind circulation. Further, Jin et al. [12] have also pointed out the increase in surface greenness and soil moisture over the North-Western Indian subcontinent thereby reducing local dust emission. It is to be noted that these studies have mostly used datasets from state-of-the-art observations and explained the potential causes for the dust reduction. However, the associated implications on regional climate and potential change in the hydrological cycle were not attempted.

The complex forcing and feedback mechanisms render elucidating changes to dust and their regional climate interaction from other large scale forcings. Further, there are no well-accepted direct measurements to distinguish dust optical and radiative properties of total aerosols from space. Hence, a model can be an optimal solution for exploring dust processes and their optical and radiative properties. The inadequate representation of various processes, coarse spatial resolution of soil data, and meteorological fields in typical global climate models (GCMs) make it difficult to study dust and its feedback at regional scales. This makes the choice of a regional climate model (RCM) better over the global models [26–29] with its higher spatial resolutions, better parameterisation, and capacity to resolve subgrid processes [28,30].

In this paper, a regional climate model (RegCM4.5) is used to understand changes to dust aerosol loading over the arid regions to the North-West of the subcontinent. The study specifically explores the relationship between rainfall, dust emission, and its atmospheric loading. In addition, the changes (quantification) in short and longwave radiation at the surface, atmosphere, and top of the atmosphere during the period are elucidated.

## 2. Data and Methodology

### 2.1. Regional Climate Model RegCM (Version 4.5)

The fourth generation of Abdus Salam International Centre for Theoretical Physics (ICTP) regional climate model (RegCM version 4.5) is used to carry out this study [31]. RegCM's dynamical core is based on the hydrostatic version of the mesoscale model MM5 of the National Center for Atmospheric Research and is widely used [32–34]. Radiative transfer in RegCM uses the parameterisation of NCAR's community climate model CCM [35]. The model's capability in simulating aerosols and their various optical properties [15,36,37] has been utilised for various applications. It also captures the mean patterns and features of other meteorological parameters (temperature, wind, precipitation, etc.) over Indian regions [34,37–40].

RegCM4.5 has both online dust and anthropogenic aerosol modules, hence it is widely used to study aerosol–climate interactions [30,36,37]. The dust module is designed to be online, i.e., it works in a two-way coupling with meteorology. This is one of the major advantages over other offline (one-way interacting) models. The online dust module follows the parameterization scheme developed by [41]. The module has major features such as dust emission, dry and wet deposition, transport, optical properties, and direct radiative forcing [42]. This dust module divides the size distribution of the dust particles into four bins varying from finer (0.01–1.0 μm and 1.0–2.5 μm) to coarser dust aerosols (2.5–5.0 μm and 5.0–20.0 μm). The model uses Mie theory to estimate the aerosol optical and radiative properties [33,42,43]. For the refractive index of dust aerosols, the model uses the Optical Properties of Aerosols and Clouds (OPAC) database [42,44].

The complexity and representation of the land model play a vital role in dust generation in RegCM. The default is the BATS (Biosphere atmosphere transfer scheme), and two community land models (CLM3.5 and CLM4.5) options are available for coupled land–atmosphere configurations. However, the benefits of CLM 4.5 are better than BATS and CLM, as the former has more soil layers, vegetation fractions, and uses a subgrid "tiles" method where a separate water and energy balance is performed for each tile [37,45]. This method aims to model the surface parameters better than the default BATS scheme. Therefore, the CLM4.5 model has been used for dust simulation in the present study.

## 2.2. Experimental Design and Dataset Used

The simulations are carried out at 50 km spatial resolution, and the initial and lateral boundary conditions are forced with 6-hourly ERA-interim 1.5° × 1.5° gridded reanalysis data. The sea surface temperature data (Weekly Optimal Interpolation dataset) are extracted from the National Oceanic and Atmospheric Administration (NOAA) 1.0° × 1.0° weekly dataset. The optimum interpolation (OI) sea surface temperature (SST) analysis is produced weekly on a one-degree grid. The analysis uses in situ and satellite SSTs plus SSTs simulated by sea ice cover. Before the analysis is computed, the satellite data are bias-adjusted. The land use data and terrain heights are generated from the United States Geographical Survey at 30s resolution. A total of 15 years (2001–2015) of simulations have been performed starting from 1st March to 31st May each year. In the present analysis, only data from May are used. Figure 1a depicts the model run domain for this study, and Figure 1b shows the various dominant soil categories associated with the land model (CLM45). This study is specially planned to investigate the associated cause and impacts of long-term dust reductions using a regional coupled chemistry climate model. The past studies could not segregate the "dust only effect" from the total aerosol. Therefore, our modelling study tried to overcome those limitations. Though there are a number of options available in the model for aerosol/chemistry boundary conditions, we have selected the "dust aerosol boundary condition" option from the name list to delineate the "dust only effects" for this study. More technical details related to dust representation, generation, and their corresponding mathematical relationships can be found in Appendix A.

The results are analyzed over the North-Western (NW) parts of India and the adjacent regions (NW box hereafter, marked as the black box on Figure 1a). The rectangular box delineated over 23.5°–33° N latitude and 66.5°–74° E longitude is our region of interest. This region is chosen as the significant deserts in the Indo-Pak (Thar, Thall, and Cholistan deserts) fall within them. This area is the primary dust source contributing to the total dust AOD load over India during the summer season. The model configurations and other details used for this study are provided in Table 1.

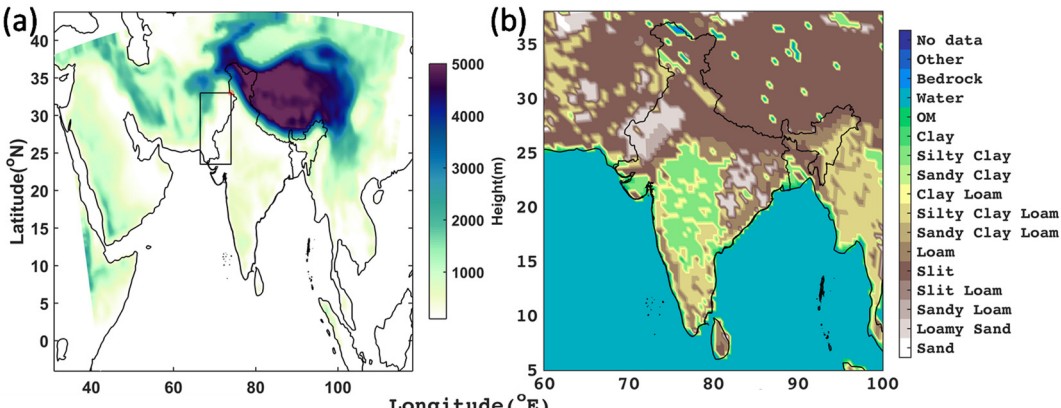

**Figure 1.** (**a**) The model domain with topographical height (shaded); (**b**) associated soil texture dominant categories. The North-Western (NW) parts of India and the adjacent regions are marked as the black box in (**a**).

**Table 1.** Model configuration implemented for this study.

| Model Used | RegCM (Version 4.5) |
|---|---|
| Grid dimensions | 160 × 100, 18 sigma levels |
| Dynamics | MM5 hydrostatic |
| Horizontal resolutions | 50 km |
| Simulation Periods | 2001–2015 |
| Top layer Pressure | 50 hPa |
| Land Surface model | CLM4.5 |
| Meteorological boundary conditions | ERA-Interim [46] |
| Chemical boundary conditions | Dust Chemistry (online) |
| Cumulus convection scheme | Emanuel over land and ocean |
| Radiation scheme | CCM3 |
| Moisture scheme | Subgrid Explicit Moisture Scheme [47] |
| Planetary boundary layer scheme | Holtslag PBL [48] |
| Topography | USGS |
| SST | Weekly Optimal Interpolation dataset (OI_WK) |
| Dust tracers | DUST4 (4 bins) |
| Dust size particle distributions | Standard scheme [42] |

## 3. Results and Discussion

This section explains the simulated spatial pattern and trends of dust AOD, rainfall, tracer burden, and radiative parameters related to dust. The results section is divided into two parts. The first part explains the spatial trends of the parameters mentioned above. In the second part, we calculate the long-term area-averaged temporal trends of these parameters over the NW box. We further discuss the implications of the changes in the above parameters to the regional climate.

### 3.1. Trends in AOD and Precipitation

The spatial trend of simulated aerosol optical depth (AOD) for May (2001–2015) is shown in Figure 2. The simulated pattern is similar to that observed using MERRA2 (Modern-Era Retrospective analysis for Research and Applications, version 2, Figure S1a,b). The AOD shows a declining trend over the whole NW part and significant parts of central Indian regions. The signal is robust, especially within the study area of interest (the dotted black box). It attains the maximum negative trends ($\sim -0.1$ year$^{-1}$) over the Thar Desert regions. However, a slightly positive trend was observed east of the area of interest, though not statistically significant ($p > 0.05$). To find the cause behind the decline in AOD trends, the simulated precipitation trend during the same period (Figure 2b) is also explored.

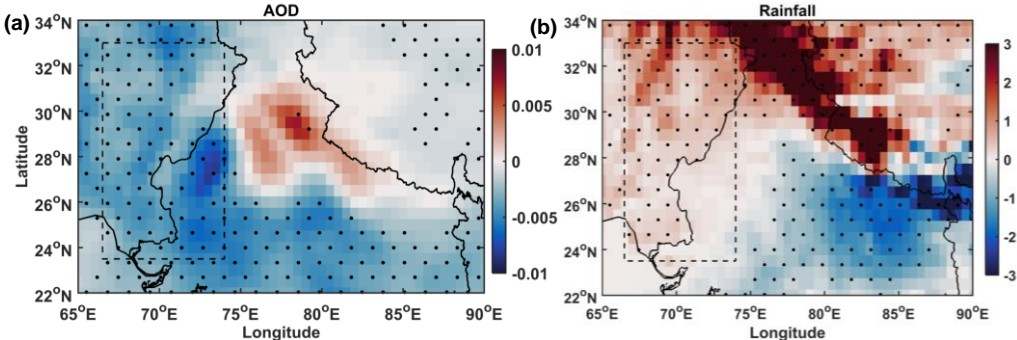

**Figure 2.** (**a**) Trends (year$^{-1}$) in AOD and (**b**) rainfall for the period of study. Stipple shows the area where trends are significant at 95% confidence level, determined using the Mann-Kendall trend tests.

The precipitation trend resembles that observed using the University of Delaware (UDel) precipitation data (Figure S1c). It shows a strong positive trend over the NW (especially over Pakistan). However, there is a sharp reduction in simulated precipitation trends covering parts of Central India and located east of the area of interest. It may be noted that the simulated trend in AOD and the precipitation over the rectangular box are in the opposite phase. The observed aerosol reduction further to the east might be due to suppression of dust transportation from the source regions (here, the area covering the rectangular box). To find the possible pathway of the dust/AOD reduction, we further investigated the associated change in the tracer (dust) emissions, its burden, and deposition. Those are discussed in the subsequent sections.

*3.2. Changes in the Dry and Wet Deposition of Dust*

The surface emission and the dust burden are primarily controlled by the wind speed and soil texture in the RegCM [31,42]. On the other hand, the dust removal from the atmosphere is controlled by dry and wet removal processes (due to gravity and rain, respectively). Dust column burden is directly linked to the surface-emission (from the surface to atmosphere), loss (due to dry deposition and wet deposition), and transport (by wind). The dust column burden follows the trend as observed for AOD (not shown). Figure 3 explains the trend of two major dust removal processes. It is clear that both the dry deposition (gravity settling) and wet deposition (washout due to rain) shows a positive trend over the NW box. The positive trend in tracer loss (combined effect of dry and wet deposition) contributes to a reduction in aerosol optical depth. In addition, the wet deposition exceeds dry deposition more than five-fold in magnitude when averaged over the NW box (see Table 2). This signifies that rain is one of the primary causes of dust removal from the atmosphere, and hence is responsible for the negative AOD trends, as observed in Figure 1.

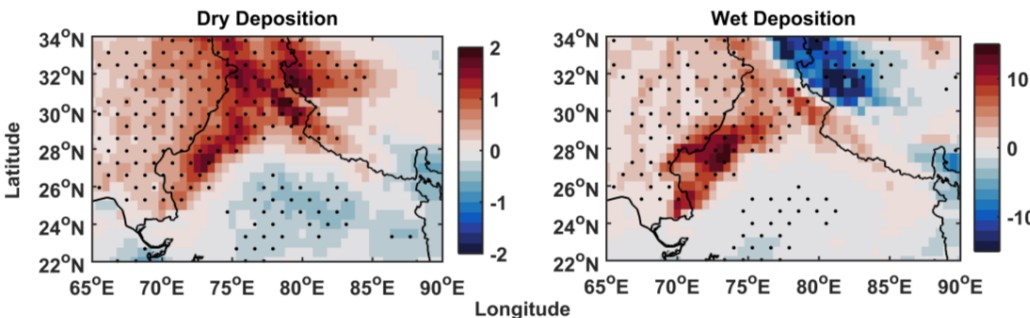

**Figure 3.** Trends (year$^{-1}$) in deposition (dry and wet, mgm$^{-2}$day$^{-1}$) for the period of study. Stipple shows the area where trends are significant at 95% confidence level, determined using the Mann-Kendall trend tests.

It is established that pre-monsoon dust contributes large fractions to annual average AOD over Indian landmass [49–52]. Previous studies suggest the dust load/AOD during April–May has a role in modulating the rainfall during monsoon months through an elevated heat pump mechanism [15,24,53]. Thus, long/short term changes in dust loads/absorbing AOD can impact regional climate and hydrological cycles. As regional climate is controlled mainly by the change in radiative balance over that region, a positive (negative) radiative balance leads to warming (cooling). It is to be noted that dust is well known to interact with both short and longwave radiation. Therefore, this decline in dust load is expected to change the regional radiative balance in near-surface (boa), top of the atmosphere (toa), and in the atmosphere (atm), the details of which are discussed in the next section.

### 3.3. Changes in Radiative Forcing and Heating Rate

The aerosol (dust) radiation interaction is known to as direct radiative forcing (DRF or RF), which is meant to study the climate implication of aerosols. It is calculated as the change in radiative fluxes (incoming minus outgoing) considering conditions with and without aerosol. The DRF is estimated at the top of the atmosphere ($RF_{toa}$) as well as the surface/bottom of the atmosphere ($RF_{boa}$). The direct atmospheric forcing ($RF_{atm}$) is the difference between radiative forcing at the top and bottom of the atmosphere as follows.

$$RF_{atm(SW,LW)} = RF_{toa(SW,LW)} - RF_{boa(SW,LW)} \tag{1}$$

In the above equation, SW/LW denotes the radiative calculation both in the shortwave or longwave radiation spectrum. Dust is known to behave differently in the shortwave (SW) and longwave (LW) spectrum of radiation. Hence, we have investigated them separately. All the units of radiative forcing are in $Wm^{-2}$.

Figure 4 shows the trend in radiative forcing (RF) simulated by the model. The left four panels of Figure 4 depict the long-term changes in shortwave radiative (toa, boa, atm) and heating rate (SWHR) due to shortwave radiation. The right panel explains the same variable in longwave spectrums of radiation. The simulated shortwave top of the atmosphere forcing (SWRFtoa) shows an increasing trend over the NW box and North-Central India. A different response is observed for longwave radiative forcing at the top of the atmosphere. It is well known that dust aerosols show both a scattering and absorbing nature at the top of the atmosphere for shortwave radiation; however, the net effect is negative (cooling). Hence, a reduction in dust aerosols enhances warming (shortwave trap) at the top of the atmosphere.

Similarly, in the longwave spectrum, dust is absorbing in nature. Therefore, the declining trends in the longwave top of the atmosphere radiative forcing (LWRFtoa) agree with the observed trends in dust burden and/or AOD. Interestingly, the longwave cooling trend exceeds the shortwave warming trends both in values and spatial extensions, leading to a net cooling effect at the top of the atmosphere. The nature of the radiative forcing at the surface for the shortwave (SWRF boa) and longwave (LWRFboa) provide distinct characteristics. The overall trend is positive for shortwave and a slightly negative for longwave. This might be due to the greater (than the average) incoming solar radiation to the surface and greater emission of longwave from the surface. For both cases, the decline in the dust is a favourable condition. It is interesting to note that the spatial extent of dust-related changes in the surface and top of the atmosphere radiative forcing have shown a spread beyond the source region. In addition, a decline in the trend net (shortwave and longwave combined) atmospheric radiative forcing (top minus bottom) is observed.

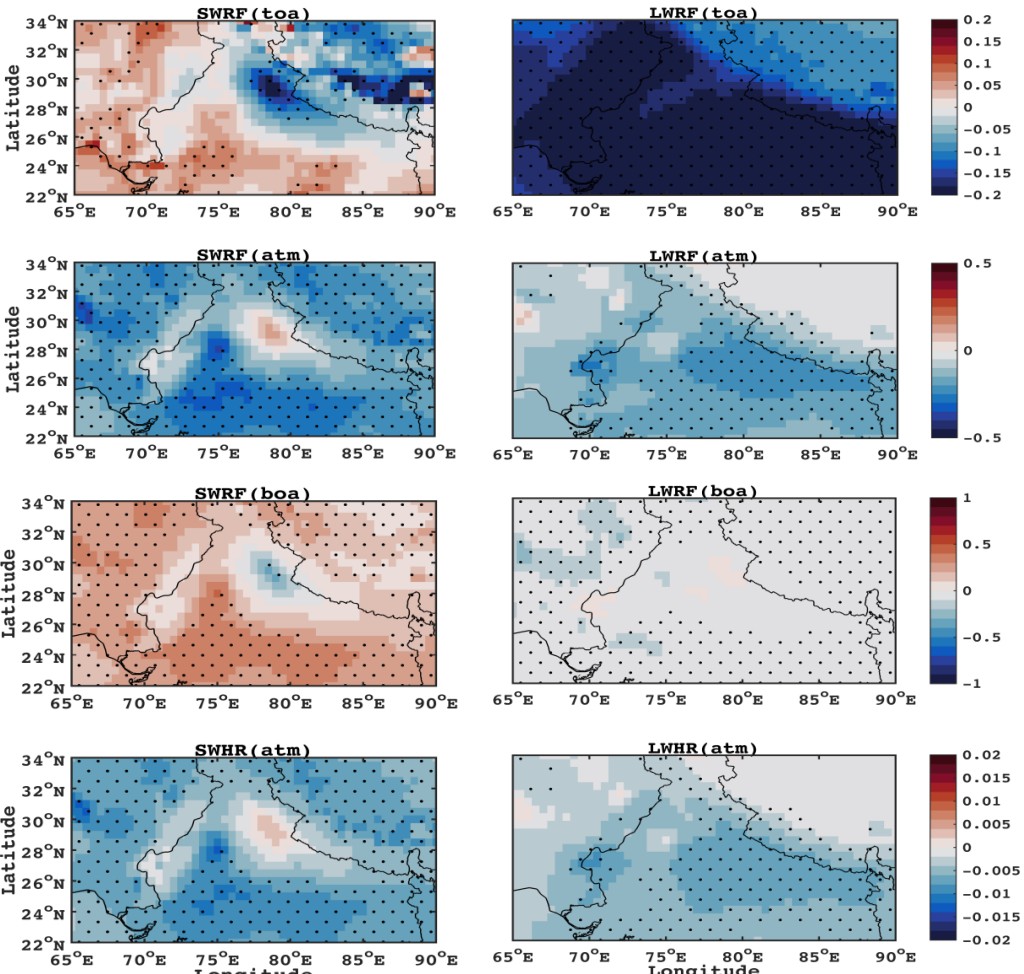

**Figure 4.** The trends (year$^{-1}$) in simulated radiative forcing (Wm$^{-2}$) and heating rate (K day$^{-1}$) in both SW and LW. The terms toa, boa, and atm are acronym for "Top of the atmosphere", "bottom of the atmosphere", and "in the atmospheric column", respectively. The stipple shows the area where trends are significant at 95% confidence level, determined using the Mann-Kendall trend test.

It may further be noted that the spatial trends of shortwave forcing are dominant mostly over the NW box and part of North-Western India. On the other hand, spatial trends of longwave forcing are prevailing over the Eastern regions. The atmospheric radiative forcing explains the amount of radiative flux absorbed or lost by the atmosphere due to the presence of an aerosol species. This is further explained using a matrix called the atmospheric heating rate (SWHR and LWHR in Figure 4) as given in Equation (2). The atmospheric heating rate is considered to be a sign of climate implication of aerosols and as calculated from RFatm [54–56]

$$\frac{\delta T}{\delta t} = \frac{g}{C_p} \frac{\Delta RF_{atm}}{\Delta P} \tag{2}$$

where g is the acceleration due to gravity, Cp the specific heat of air at constant pressure, and P is the atmospheric pressure. ΔP is the atmospheric pressure difference between the top and bottom boundary of each layer, respectively. As most of the aerosol load (here dust), thereby heating, is confined to the lower atmosphere, a constant value of 300 hPa is used for ΔP in Equation (2) [54,55].

From Figure 4, it is clear that the simulated heating rate (both for the shortwave and longwave) shows a declining trend with an agreement with the aerosol burden and/or atmospheric radiative forcing. It has a more extensive spatio-temporal spread beyond the

source region. The negative trend of the shortwave exceeds the longwave in magnitude. This is attributable to reducing dust, hence the change in atmospheric absorption in both the short and longwave radiation spectrum. The next section quantifies the net changes in dust and associated parameters over the selected region (i.e., NW box).

### 3.4. Trends in AOD and Precipitation over the North West

Figure 5 depicts the inter-annual variation of the simulated rainfall and AOD, averaged over the NW box, specified in Figure 2. Please note that the trend has been calculated taking anomalies for the month of May of different years with respect to the year 2001 (the first year of simulation or base year hereafter). It is visible that the rainfall and AOD during the summer (month of May) exhibit opposite trends.

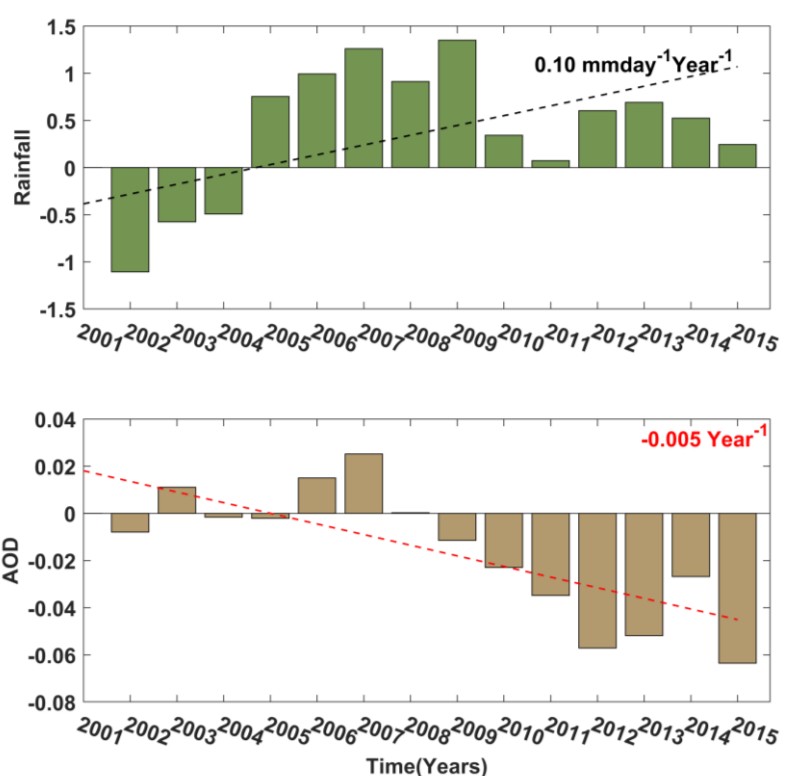

**Figure 5.** Trends (year$^{-1}$) within NW box with respect to the base year (2001) in simulated dust AOD and precipitation for the period of study over the region of interest (NW part of India and adjacent regions). Trends are significant at the 95% confidence level determined using the Mann-Kendall trend tests.

The rainfall (top panel of Figure 5) shows a significant positive trend (0.1 mm day$^{-1}$year$^{-1}$, $p < 0.05$) with respect to the base year of simulation. On the other hand, AOD in the bottom panel shows a declining trend over the years (0.005 Year$^{-1}$, $p < 0.05$). These results show evidence that the atmosphere over the dust source region is becoming cleaner during the recent decade as a response to changes in rainfall. Therefore, further changes in rainfall over this region are of importance to regional aerosol loading. The current study thus supports the findings by Pandey et al. [25]. In addition, this study quantifies the change in aerosol radiative effects concerning the overall dust change. This also points to the possibility of using such aerosol-chemistry models for studies related to regional hydrology. Again, the increased rainfall over the arid/semi-arid region indicates a change in the regional climate as dust simulated by the model simply responds to the large-scale meteorological fields.

### 3.5. Trends in Dust and the Associated Changes in Radiative Forcing

We estimate the trends and net changes in the dust, rain, and radiative forcing due to shortwave and longwave spectrum (see Table 2) over NW. The inter-annual variation for each parameter is shown in Figure S2. It may be noted that the anomalies were calculated using a similar approach, as discussed in Section 3.4.

**Table 2.** Area averaged trends of various variables with respect to year 2001.

| Variables | Units | Trend Year$^{-1}$ | Duration | Total Change (%) |
|---|---|---|---|---|
| AOD | unit less | **−0.005** | 2001–2015 | −17.5% |
| Precipitation | mm day$^{-1}$ | **0.10** | 2001–2015 | 54.14% |
| Burden | mg m$^{-2}$ | **−5.47** | 2001–2015 | −19.3% |
| Surface Emission | mg m$^{-2}$ day$^{-1}$ | −8.38 | 2001–2015 | −3.6% |
| Dry Deposition | mg m$^{-2}$ day$^{-1}$ | **0.28** | 2001–2015 | 48.6% |
| Wet deposition | mg m$^{-2}$ day$^{-1}$ | **1.69** | 2001–2015 | 79.6% |
| SWRF (toa) | Wm$^{-2}$ | **0.04** | 2001–2015 | 7% |
| SWRF (atm) | Wm$^{-2}$ | **−0.19** | 2001–2015 | −16.6% |
| SWRF (boa) | Wm$^{-2}$ | **0.24** | 2001–2015 | 11.6% |
| SWHR (atm) | Kday$^{-1}$ | **−0.006** | 2001–2015 | −16.1% |
| LWRF (toa) | Wm$^{-2}$ | **−0.14** | 2001–2015 | −88.6% |
| LWRF (atm) | Wm$^{-2}$ | −0.08 | 2001–2015 | −32.8% |
| LWRF (boa) | Wm$^{-2}$ | **−0.09** | 2001–2015 | −20.8% |
| LWHR (atm) | Kday$^{-1}$ | −0.002 | 2001–2015 | −22.5% |

**Bold numbers** indicate statistically significant trends ($p < 0.05$).

Similar to the spatial trends, dust tracer burden and surface emission both exhibit negative trends during the summer season (Month of May). This may be due to the observed increase in rainfall, which makes the atmosphere relatively cleaner, and at the same time, the land surface becomes wetter. A wetter land inhibits soil erosion and dust emission. Further, the increase in wet depositions supports the precipitation link to dust reduction. On the other hand, an increase in dry deposition may be attributed to the dust particle growth in the presence of additional moisture. Hence, the combined effects of dry and wet deposition and the rainfall might be responsible for the trends of simulated dust aerosol optical depth. The area-averaged temporal trends of shortwave and longwave radiative elements agree with the change in dust load/AOD. It is essential to note that these radiative elements are linked to the state of the climate of a region. Long-term changes in any of these variables indicate a change in the regional climate. There are notable alterations observed in the radiative elements during the study period. Please note that even though year to year monthly variations are observed, we have discussed only the mean changes of these elements during summer (May) for the study period. It may be mentioned that regional climate models are meant to simulate the mean and variability, rather than exact quantification of a climate state variable. Area averaged temporal trends in shortwave radiative forcing show robust and significant trends ($p < 0.05$) over its longwave counterparts. The trends in both TOA and BOA radiative forcing components are as expected. Dust aerosols are known to warm the atmosphere by absorbing the incident solar radiation, and this can be quantitatively determined by the atmospheric heating rate. The decrease in aerosol-induced atmospheric heating over south Asia has also been reported in a recent study [57].

The overall changes in the above-discussed parameters over the NW box are expressed in percentage (Figure 6). A total change in the AOD with respect to the base year is estimated to be −17% and results in a 19% decrease in dust burden. The direction and magnitude of the change in burden and AOD are comparable with earlier studies [25,52]. The precipitation has increased more than 50% over the semi-arid/desert regions. In addition to this, wet deposition percentage change (~80%) exceeds almost twice that of dry depositions change (~48%).

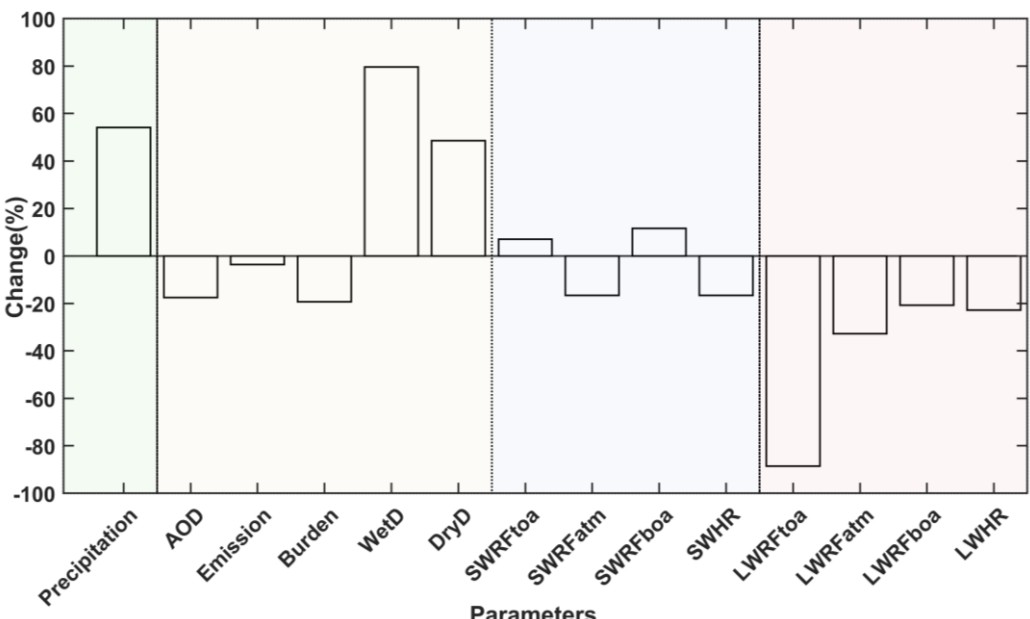

**Figure 6.** Overall changes (percentage) in various parameters within the NW box with respect to the base year (2001). SW, LW, and HR are acronyms for shortwave, longwave, and heating rate, respectively.

There are net increases in the top and bottom of the atmosphere shortwave radiative forcing (7 and 11.6%, respectively). However, there is an overall decline in the atmospheric heating and atmospheric radiative forcing (~16%) in the shortwave domain. Although the shortwave radiative forcing and heating rate trends are much higher than those of its longwave counterparts, the change in longwave forcing is more significant in magnitude and mostly negative, as shown in Figure 6. Further, maximum change is observed in top of the atmosphere LWRF and is approximately −88% from that of the base year. It is interesting to note that the LWRFatm responds nearly twice over SWRFatm to a unit change in dust AOD/burden. These results are essential to discern the response of shortwave and longwave forcing to the same factor (here the change in dust).

## 4. Conclusions

Regionally, dust aerosols partly offset greenhouse gas (GHGs) warming due to inhibiting shortwave radiation from reaching the ground. Further, by warming the atmospheric column, it is linked to various boundary layer processes. In both short- and long-time scales, dust aerosols impact the monsoon rainfall over India, as suggested by previous studies [9,11,58]. Hence, any change in dust magnitude/spread could affect the regional climate through various forcing and feedback mechanisms. Using a coupled chemistry-climate model with an updated land module (RegCM4.5), we simulated the summer changes in dust over the source regions adjacent to Indian landmass for 15 years (2001–2015) period span. The model performance is reasonable compared to the observations and is in fair agreement with the findings of earlier studies. Additionally, ground-based atmospheric aerosol particle sizing with real-time particle sizing for ultrafine to the coarse particles at various locations may give more information about these dust particles' characteristics. In summary:

1. The simulated AOD is found to exhibit a spatio-temporal declining trend, whereas rainfall shows an increasing trend over the arid/desert and semi-arid regions of NW India.
2. The change in AOD could be due to the combined effects of rainfall and tracer processes (emission, transport, and removal/deposition). There is an approximately

18% decline in AOD and >50% rainfall enhancement observed over NW parts of India.

3.  The trend of wet removal is more than five-fold more dominant over the gravity settling/dry deposition. However, the net change in wet removal is nearly twice that of dry deposition.
4.  The direction of the observed trend in radiative forcing in both shortwave and long-wave radiation regimes agree with the change in dust load/burden. However, the shortwave forcing trends are dominant over its longwave counterparts.
5.  Longwave radiative forcing is more sensitive to a unit change in dust burden/AOD compared to shortwave radiative forcing.
6.  As a response to dust change, a significant widespread atmospheric cooling trend is observed over parts of North and North-Western India.

**Supplementary Materials:** The following are available online at https://www.mdpi.com/article/10.3390/rs13214309/s1, Figure S1: Trends for pre-monsoon dust AOD, Dust burden, and precipitation from MERRA-2 and Udel, Figure S2: Trends within the NW box with respect to the base year (2001) in various parameters with the corresponding sign.

**Author Contributions:** A.A., S.K.P., and V.V. designed the research; A.A. simulated the model, analysed the data, and wrote the manuscript; A.A., S.K.P., V.V., R.R., and N.M. revised the manuscript. All authors have read and agreed to the published version of the manuscript.

**Funding:** This research received no external funding.

**Institutional Review Board Statement:** Not applicable.

**Informed Consent Statement:** Not applicable.

**Data Availability Statement:** All the ground-based observational data used in this study can be obtained from the corresponding author upon request.

**Acknowledgments:** A.A. is thankful to the Department of Science and Technology Government of India for providing INSPIRE fellowship for doctoral research. We are thankful to the International Centre for Theoretical Physics (ICTP) for providing the RegCM 4.5 source code. We are also grateful to the developers from Max Planck Institute für Meteorologie and National Center for Atmospheric Research (NCAR) for providing open-source software such as CDO and NCL, which are used in this study. We are also thankful to TSI (TSI Instruments India Private Limited, Bangalore, India) for providing technical support during the preparation of this manuscript. V.V. thanks ISRO for support through its ARFINET program. IIT Bhubaneswar is acknowledged for providing the necessary infrastructure while this research was carried out.

**Conflicts of Interest:** The authors declare no conflict of interest.

## Appendix A

The RegCM dust module has been proven to have a good capability of modelling the spatial distribution of surface dust concentration variations. In this study, using RegCM version 4.5, we have documented the long-term dust changes over the northwestern parts of Indian and adjacent desert regions. We also investigated the possible cause behind the dust change and associated changes/effects to atmospheric radiation and thermodynamics.

Dust simulation by RegCM 4.5.

Dust emission in any model primarily depends on the land use, soil types, erodibility, and meteorological conditions. To represent the land module, we have coupled the model with CLM4.5 over the default BATS (more can be found in the Methods section).

Dust flux is the main parameter in estimating the burden of aeolian dust. We activated the 4 dust bins scheme for aerosol options. To activate that, one has to opt for the "*chemsimtype* to *DUST*" in the "*chemparam*" name list. In the case of a DUST simulation, we need the model to prepare a soil type dataset to be used to calculate the dust emission. To do that, one has to opt for the *ltexture* in the *terrainparam* name list to be *true*.

The size bins are between 0.01 and 1 μm, 1 and 2.5 μm, 2.5 and 5 μm, and 5 and 20 μm for DUST 1, DUST 2, DUST 3, and DUST 4, respectively. The dust emission size distributions are calculated according to Zakey et al. 2006 [42]. The size dust distribution from Alfaro and Gomes is used [41]

To calculate the dust emission in RegCM, the following steps and calculations are considered.

(a)Specification of soil aggregate size distribution for each grid cell, (b) calculation of threshold friction velocity, (c) calculation of horizontal saltation soil aggregate mass flux, and (d) calculation of vertical transportable dust mass flux.

Calculation of the horizontal saltating mass flux $dH_F(D_P)$ and calculation for a saltating aggregate of size $D_p$ is primarily from Marticorena and Bergametti [59] and shown in Equation (A1).

$$dH_F(D_P) = E\frac{\rho_a}{g}u^{*3}(1 + R(D_P))\left(1 - R^2(D_P)\right)dS_{rel}(D_P) \tag{A1}$$

where $E$ is the ratio of the erodible surface to total surface, $\rho_a$ is the air density, $g$ is the gravity, $u^*$ is the wind friction velocity calculated for each grid cell. $R(D_P)$ is the ratio of the threshold friction velocity to the friction velocity, and $dS_{\rm rel}(D_P)$ is the relative surface of soil aggregate of diameter $D_p$ to the total aggregate surface.

The vertical mass flux of transportable dust particles is calculated according to Equation (A2)

$$F_{{\rm dust},i}(D_P) = \frac{\pi}{6}\rho_p D_i^3 N_t \tag{A2}$$

where $(D_P)$ is the diameter of the particle, $D_i$ is the median diameter of $i$ th mode, and $\rho_p$ is the particle density. The $N_i$ is calculated according to Equation (A3) as follows

$$dN_i(D_p) = dF_{\rm kin}(D_p)p_iD_p/e_i \tag{A3}$$

and

$$dF_{\rm kin}(D_p) = \beta dH_F(D_P) \tag{A4}$$

The $p_iD_p$ is the fraction of the kinetic energy of the saltating aggregate used to release dust particles in the $i$th emission mode, and $e_i$ is the binding energy attached to the $i$th emission mode. The $\beta$ is a constant and as is approximately 16,300 cm/s$^2$ [60].

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
