# Peer review of "Assessment of Recent Changes in Dust over South Asia Using RegCM4 Regional Climate Model"

_remotesensing, doi:10.3390/rs13214309_

Round 1

Reviewer 1 Report

General comments:

Asutosh et al. used a regional climate model with an updated land module, and simulated the long term changes of dust over south Asia. The study is generally interesting and well-written. Minor revision is necessary prior to publication of the study.

Specific comments:

L21: “day-1 year-1”: here and throughout the manuscript use superscripts correctly

L114: why only data for May were used?

L194: “as” instead of “to be”

Figure 4: Please add units to the colorbars

L215: “depict” instead of “depicts”

L230: “slightly” instead of “a slight”

L232: “the surface” instead of “a surface”

Figure 5: Although the authors have calculated average linear trends, one can tell from the graphs that rainfall increased in 2001 – 2007, and then decreased, while the decrease in AOD is stronger after 2008. I believe that the authors should comment on this.

L227: “provide” instead of “show”

L294: I suppose that “w.r.” means “with respect”

L309: “Though … observed”: please rephrase

L323: “earlier” instead of “the earlier”

L320 – 321: “The … percentage”: please rephrase because the meaning is not clear

L330: Delete “)”

L336: “Although” instead of “Though”?

L360: Delete “to be”

Reviewer 2 Report

Overall, the study is interesting, but confusing. There are a lot of topics that were not clear to me, as well as the reason why you did some analyzes. I could agree to accept the paper under some revision on its content.

Introduction: I missed more literate on your Introduction to reinforce the reason why you have decided to do this particularly study. If possible, please add more information.

Methodology: As in the introduction, it requires more information about your runs:

-why did you decide to use dust chemistry boundary conditions? Have you tried to use another one?

Maybe a more detailed description about your boundary conditions would clarify some of your results/runs.

How did you separate the dust portion from the total AOD? This is unclear.

Results: I found your description of your results kind of confusing. I also miss more discussion about what you’ve found.

Reviewer 3 Report

Please see comments in the attached PDF

Round 2

Reviewer 2 Report

Thank you for the improvements on your paper and for considering my notes.

Overall the text is much more clear than it was before. I juts found some minor coorections regarding your references. They are stated above.

Line 60/61: Please check the reference style. The same goes for lines 64/65.

In my opinion the paper can be accepted in the current form.

Best luck!